# Donor Impurity in CdS/ZnS Spherical Quantum Dots under Applied Electric and Magnetic Fields

**DOI:** 10.3390/nano12224014

**Published:** 2022-11-15

**Authors:** Kobra Hasanirokh, Adrian Radu, Carlos A. Duque

**Affiliations:** 1Research Institute for Applied Physics & Astronomy, University of Tabriz, Tabriz 51665-163, Iran; 2Department of Physics, Politehnica University of Bucharest, Splaiul Independenței 313, RO-060042 Bucharest, Romania; 3Grupo de Materia Condensada-UdeA, Instituto de Física, Facultad de Ciencias Exactas y Naturales, Universidad de Antioquia UdeA, Calle 70 No. 52-21, Medellín AA 1226, Colombia

**Keywords:** core/shell quantum dot, donor impurity, binding energy, electric field, magnetic field, nonlinear optical properties

## Abstract

This article presents a theoretical study of the electronic, impurity-related, and nonlinear optical properties of CdS/ZnS quantum dots subjected to electric and magnetic fields. The magnetic field is applied along the *z*-axis, with the donor impurity always located in the center of the quantum dot. In the case of the electric field, two situations have been considered: applied along the *z*-axis and applied in the radial direction (central electric field). In both cases, the azimuthal symmetry (around the *z*-axis) is preserved. In the absence of a magnetic field and considering a central electric field, the system preserves its spherical symmetry both in the presence and in the absence of the donor impurity. The study is carried out in the effective mass approximation and it uses the finite element method to find the eigenfunctions and their corresponding energies, both in the presence and in the absence of the impurity. This work investigates the optical absorption coefficient and the relative change of the refractive index, considering only intraband transitions between *l* = 0 states (states with azimuthal symmetry concerning the *z*-axis). Calculations are for *z*-polarized incident radiation. The study shows that the combined effects of a central electric field and a *z*-directed magnetic field can give rise to a typical core/shell-like quantum confinement with oscillations of the electron ground state. Additionally, it is shown that the presence of the donor impurity suppresses such oscillations and it is responsible for blue shifts in the optical properties and magnifications of the corresponding resonances.

## 1. Introduction

The charge carriers’ motion in quantum dots (QDs) is confined in three spatial directions, so these nanostructures are very significant for processes favored by strong confinements. The semiconductor QDs have attracted significant attention due to their device applications [1,2,3,4], mainly to those related with the electro-optical properties. The synthesis methods, such as molecular beam epitaxy and metal organic chemical vapor deposition, allow the experimentalist to grow semiconductor QDs with different geometries and compositions. The typical core–shell QD consists of a smaller band gap QD, known as the core, that is covered by another semiconductor material with a larger band gap, or the shell. The core-shell QDs are composed of II–VI, III–V, or IV–IV semiconductors and their physical properties strongly depends on the band gaps [5]. The nonlinear optical properties of semiconductor QDs have attracted great attention both for experimental and theoretical studies [6,7,8,9,10,11,12,13,14]. The results of these studies enable a broad range of potential optoelectronic devices. In the presence of impurities and/or various external factors, it takes remarkable theoretical effort to calculate the energy spectrum and wavefunctions and it is generally not feasible to find analytical solutions. Due to rather complicated practical geometries, boundary conditions, and surrounding environment effects, researchers have resorted to different approximation methods [15,16,17].

The results found for CdS films chemically deposited from solutions containing different impurities, known as donor-type dopants for CdS (chlorine, iodine, boron, and indium), have been reported by Altosaar et al. [18]. In CdS, Cd and S vacancies result in the creation of various acceptor and donor states within the band gap. This will cause the transition of the charge carriers through impurity states to the valence band, followed by the broadening of the photoluminescence spectra, as well as the deviation from the ideal band gap energy of the pure semiconductor [19]. When fluorine atoms are incorporated into the CdS lattice, they can induce donor impurity levels within the CdS band gap; these levels will eventually produce a CdS band gap reduction for an increased fluorine concentration in the chemical bath [20].

The diagonalization method is an effective approach for studying the energy spectrum in the presence of the magnetic field (MF), and the electron localization in multilayer spherical QDs [21,22,23,24,25,26,27]. Mkrtchyan et al. have reported a theoretical study of the external MF effects on the interband and intraband optical properties of an asymmetric biconvex lens-shaped QD [28]. In their model, the authors resort to an adiabatic approximation allowing them to obtain analytical solutions for the eigenvalue equations, in terms of sinusoidal and confluent hypergeometric functions. Leaving aside the effects of the electron–hole interaction, the authors report the interband absorption coefficient, the photoluminescence coefficient, and the first- and third-order corrections of the optical absorption coefficient. In the presence of a central impurity, the problem can be solved by using Mathieu and Whittaker functions, Coulomb wave functions, and degenerate hypergeometric functions [29,30,31,32,33].

Once the structure of energy levels for the confined charge carriers (electrons or holes) within the QD region is known, the transition matrix elements can be calculated. These will further allow calculations of the optical properties related with inter-level transitions. Studies on QDs show that the presence of an MF can reduce the effective width of the confining potential. Thus, it changes the electronic states and the dipole moment [24,34]. To enhance the sensitivity of the optical properties of a nanosystem to the electric and magnetic fields, it is required to properly choose its size [35]. The electro-optic effect consists of a change in the refractive indices of materials to which a static or a low-frequency external electric field (EF) is applied. Some theoretical studies on optical properties in spherical core/shell QDs have taken into account the effects of LO-phonon interaction [36] and dielectric mismatch [37]. The influence of hydrostatic pressure on optical properties in AlAs/GaAs spherical QDs [38,39] has been investigated. The confinement models for QDs with finite and infinite potential barriers have been studied [40,41]. The studies reported in the literature have thus demonstrated that external fields are excellent tools for obtaining red or blue shifts of the optical properties which, in turn, can be useful to control the response of optoelectronic devices. The optical absorption coefficients are enhanced for QD systems under the effects of externally applied electric and magnetic fields and in the presence of shallow donor impurities. QDs of different shapes, structures, and compositions have been the most studied nanosystems. Suitable designs of the heterostructures effectively enrich the optical properties of multilayer semiconductors [42]. Semiconductor QDs with various properties are interesting candidates for many optoelectronic applications [43,44,45,46,47,48,49].

The present study deals with the optical properties related to transitions from the ground state to several excited states in CdS/ZnS spherical QDs under the electric and magnetic fields and in the presence of on-center donor impurities. Despite the particular choice of CdS/ZnS QDs, the results discussed here may be valid for other material combinations that give rise to zero-dimensional structures with spherical symmetry, for example GaAs/AlGaAs and InAs/GaAs QDs. Both the EF and MF, to be chosen along the *z*-axis, break the spherical symmetry of the problem while the version of the EF directed radially from the QD center or toward the QD center will preserve such symmetry, even in the presence of a donor impurity located in the center of the system. Technologically, it is rather complicated to think about radial EFs in QDs; however, the study proposed here can be successfully used to simulate spherical structures with materials where the built-in EFs (induced by the discontinuities of the spontaneous and piezoelectric polarizations at the boundaries of the QD structure) are present. The study was performed in the effective mass and parabolic conduction band approximations. The effects of the effective mass and dielectric constant mismatch between the dot and the barrier regions were included. Electron–impurity transition energies, dipole matrix elements, optical absorption, and refraction index coefficients were investigated using the finite elements method (FEM). As a result of this research, we show that the optical properties of CdS/ZnS spherical QDs are strongly dependent on external EFs and MF, impurities, and QD geometric parameters. The organization of the paper is as follows: Section 2 contains the presentation of the theoretical framework; in Section 3, we discuss the obtained results, and in Section 4, the conclusions are given.

## 2. Theoretical Model

An illustrative scheme of the CdS/ZnS spherical QD under study is shown in Figure 1. The inner and outer QD radii are $R_{1}$ and $R_{2}$, respectively. In the lowest part of the figure, the *r*-dependent confinement potential is represented. The potential is zero in CdS, $V_{0}$ in ZnS, and infinite in the vacuum region. The *z*-directed applied MF ($\vec{B}$) and EF (${\vec{F}}_{z}$) are indicated, together with the central EF (${\vec{F}}_{r}$).

Using the effective mass and parabolic bands approximations, the Schrödinger equation for a confined electron in a CdS/ZnS spherical QD under $\vec{B}$, ${\vec{F}}_{z}$, and ${\vec{F}}_{r}$, and in the presence of an on-center donor impurity is given by:(1)$$\left[\frac{1}{2\;m^{*,c}}{\left(\hat{p}+e\;\vec{A}\right)}^{2}+\eta_{1}\;e\;F_{z}\;z+\eta_{2}\;e\;F_{r}\;r+V\left(x,y,z\right)-\frac{\kappa \;e^{2}}{4\;\pi \;\varepsilon_{0}\;\varepsilon_{r}\;r}\right]\;\psi \left(x,y,z\right)=E\;\psi \left(x,y,z\right)\;,$$
where $r=\sqrt{x^{2}+y^{2}+z^{2}}$ is the electron–impurity distance, $\hat{p}=-i\;\hbar \;\vec{\nabla }$, m*^,c^ is the electron effective mass (c=w/b indicates the dot/barrier material), *e* is the elementary charge, and κ=0 or κ=1 disables or enables, respectively, the impurity effects. Additionally, $\vec{A}=-\frac{B}{2}\;\left(y\;\hat{i}-x\;\hat{j}\right)$ is the vector potential, where $\vec{B}=\vec{\nabla }\times \vec{A}$ comes from the symmetric gauge. V(x,y,z) is the structural potential, which is zero in the dot region and $V_{0}$ in the outer material. Additionally, $\varepsilon_{0}=8.85\times 10^{-12}$ C${}^{2}$/(Nm${}^{2}$) and $\varepsilon_{r}$ represent the QD static dielectric constant values. The $\eta_{1}$ and $\eta_{2}$ parameters are responsible for turning on (=1) or turning off (=0) the ${\vec{F}}_{z}$ and ${\vec{F}}_{r}$ EFs, whose intensities are $F_{z}$ and $F_{r}$, respectively.

Considering the azimuthal symmetry of the problem with respect to the *z*-axis, the complete electron wave function (WF) can be written in cylindrical coordinates as $\psi \left(x,y,z\right)=\psi \left(\rho ,\varphi ,z\right)=R\left(\rho ,z\right)\;e^{i\;l\;\varphi }$, where *l* is an integer number. Equation (Equation 1) reduces to
(2)$$\left[-\frac{\hbar^{2}}{2\;m^{*,c}}\nabla_{\rho ,z}^{2}+\frac{e\;\hbar \;B\;l}{2\;m^{*,c}}+\frac{e^{2}\;B^{2}\;\rho^{2}}{8\;m^{*,c}}+\frac{\hbar^{2}\;l^{2}}{2\;m^{*,c}\;\rho^{2}}+\eta_{1}\;e\;F_{z}\;z+\eta_{2}\;e\;F_{r}\;r+V\left(\rho ,z\right)-\frac{\kappa \;e^{2}}{4\;\pi \;\varepsilon_{0}\;\varepsilon_{r}\;r}\right]R\left(\rho ,z\right)=E\;R\left(\rho ,z\right)\;,$$
where $\nabla_{\rho ,z}^{2}$ is the ρ- and *z*-dependent 2D Laplacian operator and $r=\sqrt{\rho^{2}+z^{2}}$. We should note that in this article we follow the same strategy as in [50].

The WFs and energies in Equation (Equation 2) have been obtained via the FEM [51,52,53,54,55]. An extremely fine, physics-controlled mesh has been used with the following mesh statistics: 11,508 triangles, 5885 mesh vertices, 366 edge elements, 7 vertex elements, 0.6184 for the minimum quality of elements, 0.9275 for the average quality of the elements, 0.1651 for the area ratio of the elements, and 1414 nm${}^{2}$ for the mesh area.

After the electronic structure computation (WFs and energy levels), we calculate the |a〉→|b〉 related inter-subband optical properties. For the linear, nonlinear, and total optical absorption coefficient we have, respectively [56]: (3)$$\begin{array}{c} \alpha^{\left(1\right)}\left(\omega \right)=\omega \;e^{2}\;\sqrt{\frac{\mu_{0}}{\varepsilon_{r}\;\varepsilon_{0}}}\;\;\frac{\rho \;\hbar \;\Gamma_{ab}\;{\left|M_{a-b}\right|}^{2}}{{\left(E_{ab}-\hbar \;\omega \right)}^{2}+{\left(\hbar \;\Gamma_{ab}\right)}^{2}}\;, \end{array}$$
and
(4)$$\alpha^{\left(3\right)}\left(\omega ,I\right)\;=\;-\;\omega \;e^{4}\;\sqrt{\frac{\mu_{0}}{\varepsilon_{r}\;\varepsilon_{0}}}\;\;\frac{2\;I}{n\;\varepsilon_{0}\;c}\;\;\frac{\rho \;\hbar \;\Gamma_{ab}\;{\left|M_{a-b}\right|}^{4}}{{\left[{\left(E_{ab}-\hbar \;\omega \right)}^{2}+{\left(\hbar \;\Gamma_{ab}\right)}^{2}\right]}^{2}}\;,$$
and
(5)$$\begin{array}{c} \alpha \left(\omega ,I\right)=\alpha^{\left(1\right)}\left(\omega \right)+\alpha^{\left(3\right)}\left(\omega ,I\right). \end{array}$$

Concerning the relative refractive index change, the linear, nonlinear, and total coefficients are [56]
(6)$$\frac{\Delta n^{\left(1\right)}\left(\omega \right)}{n}=\frac{\rho \;e^{2}\;{\left|M_{a-b}\right|}^{2}}{2\;\varepsilon_{0}\;\varepsilon_{r}}\;\;\frac{E_{ab}-\hbar \;\omega }{{\left(E_{ab}-\hbar \;\omega \right)}^{2}+{\left(\hbar \;\Gamma_{ab}\right)}^{2}}\;,$$
(7)$$\frac{\Delta n^{\left(3\right)}\left(\omega \right)}{n}=-\;\;\;\frac{\rho \;\mu_{0}\;c\;I\;e^{4}\;{\left|M_{a-b}\right|}^{4}}{n^{3}\;\varepsilon_{0}}\;\frac{E_{ab}-\hbar \;\omega }{{\left[{\left(E_{ab}-\hbar \;\omega \right)}^{2}+{\left(\hbar \;\Gamma_{ab}\right)}^{2}\right]}^{2}}\;,$$
and
(8)$$\frac{\Delta n(\omega ,I)}{n}=\frac{\Delta n^{\left(1\right)}\left(\omega \right)}{n}+\frac{\Delta n^{\left(3\right)}\left(\omega ,I\right)}{n}\;.$$

In these expressions, $M_{a-b}=\langle \psi_{a}\left(x,y,z\right)|z|\psi_{b}\left(x,y,z\right)\rangle$ represents the inter-subband transition matrix element (the dipole matrix element divided by the electron charge), $E_{ab}=E_{b}-E_{a}$ is the main inter-subband optical transition, $\hbar \;\Gamma_{ab}=0.7$ meV is the damping term, $n=\sqrt{\varepsilon_{r}}$ is the refraction index, $\mu_{0}=4\;\pi \times 10^{-7}$ T m A${}^{-1}$ is the vacuum magnetic permeability, and I=30 MW/m${}^{2}$ is the intensity of the incident radiation. Furthermore, $\rho =3.0\times 10^{22}$ m${}^{-3}$.

We have to clarify that in Equations (4) and (7) for the third-order corrections we have omitted the terms associated with the diagonal matrix elements, $M_{aa}$, which only appear in the problems where the axial symmetry is broken. In this study, such a situation is only present in cases where EFs along the *z*-axis are considered. In general, in this report, we will only present results for the optical properties related to situations in which the azimuthal symmetry is preserved, which are achieved with MF along the *z*-axis and central EF. This choice does not affect the relevance of the research and it allows us to concentrate on the most important physical aspects of the problem.

The parameters used in this work are: $m^{*,\mathrm{CdS}}=0.19\;m_{0}$, $m^{*,\mathrm{ZnS}}=0.25\;m_{0}$, $\varepsilon^{\mathrm{CdS}}=8.3$, $\varepsilon^{\mathrm{ZnS}}=8.9$, and $V_{0}=0.8$ eV. Here, $m_{0}$ is the free electron mass.

We highlight that the numerical results presented below are valid particularly for CdS/ZnS QDs, but we remind the reader that the physics behind this is more general and can be extended to other combinations of materials. Changes in the materials essentially imply changes in the effective mass parameters, and consequently in the effective Bohr Radius and Rydberg energy, which finally translates into changes in the energies of the confined states, in the dipole matrix elements for intraband transitions, and in the impurity-related binding energies.

## 3. Results and Discussion

In Figure 2, we present the lowest confined electron states in a CdS/ZnS spherical QD as a function of the $R_{1}$-inner radius. Results are without, Figure 2a, and with, Figure 2b, on-center donor impurity with $R_{2}=30$ nm and considering several values of the *l*-quantum number, without EF and MF effects. Figure 2a shows the decreasing character of all energy levels as the internal radius of the QD increases. This effect is associated with the decrease in confinement which, consistent with the uncertainty principle, is directly connected to a drop in kinetic energy. In the large value regime of the $R_{1}$-radius, all the states shown must converge towards the energy spectrum of an electron confined in a CdS QD of radius 30 nm and infinite confinement potential. Making k=1 in Equation (Equation 2) turns on the effect of the impurity, which has an attractive character on the electron. In this case, the energy levels shown in Figure 2a must present a shift towards lower energies, as shown in Figure 2b. Considering that the expected value of the electron–impurity distance takes its minimum value for the ground state may explain why, of all the states presented in Figure 2b, the one that undergoes a greater shift towards lower energies corresponds to the first solution for l=0. Subtracting the ground state energy without and with impurity gives the ground state binding energy. Considering the results in Figure 2a,b, it is then obtained that for $R_{1}=10$ nm the binding energy is 54.7 meV while for $R_{1}=25$ nm the binding energy is 40.5 meV. It is clearly observed that this variation in the binding energy is fundamentally due to the increase in the QD radius. The effects of the variation of the QD radius on the Coulomb interaction are negligible, less than 1 meV, since the structures shown oversize an effective Bohr radius. Given the spherical symmetry of the problem at hand, in the absence or presence of a donor impurity, the energy spectrum shown in Figure 2a,b must have the same degeneracies as a hydrogenic atom in the bulk material. For example, the first excited state must be triply degenerate with the three solutions coming from the second solution for l=0 and the first solutions for both l=+1 and l=−1. The second excited state must have a fifth-order degeneracy with the states coming from the third solution for l=0, the second solution for both l=+1 and l=−1, and the first solution for l=+2 and l=−2.

Figure 3 shows the results for the lowest confined electron states in a CdS/ZnS spherical QD as functions of the central, Figure 3a,b, and *z*-directed, Figure 3c,d, EF. Results are without (left-hand column panels) and with (right-hand column panels) on-center donor impurity with $R_{1}=20$ nm and $R_{2}=30$ nm and considering several values of the *l*-quantum number, without MF effects. In Figure 3a, the positive central EF pushes the electron towards the central region of the QD, which means a greater confinement, since the volume of the space where the electronic WF is distributed decreases. This explains the increasing and positive energies for $F_{r}>0$. In the case of a negative central EF, the electronic WF is pushed towards the outer boundary of the CdS material, distributing the electronic probability in a larger space region, which implies a drop in energies, as seen for all curves in Figure 3a for $F_{r}>0$. The negative energies in Figure 3a are explained by the fact that the negative central EF has the effect of displacing the bottom of the potential well towards negative energies, particularly in the region of the interface between the QD barrier materials. The presence of the on-center impurity, combined with the central EF, presented in Figure 3b, shows once again that the ground state with l=0 is the one that reports the greatest electron–impurity interaction effects, with a strong variation in the binding energy between 17 meV for $F_{r}=-30$ kV/cm and 55 meV for $F_{r}=30$ kV/cm. Clearly, the greatest changes in binding energy occur in the range of positive EFs, which is where $F_{r}$ contributes to reducing the average electron–impurity distance, thus producing a reinforcement in the electrostatic interaction. Due to the spherical symmetry of the central EF, in Figure 3a,b the same state degeneracies that were discussed in Figure 2 are preserved. Turning now to the *z*-directed EF problem, Figure 3c,d, it is observed that a symmetric drop in energies can even lead to negative energies in the high *z*-directed EF regime. Although the *z*-directed EF compresses the WF towards the regions with $z=\pm R_{1}$ (depending of its sign) and thereby produces a confinement effect, the shifts towards lower energies of the spectrum are due to the shift towards negative energies of the bottom of the confinement potential. The presence of the impurity combined with the *z*-directed EF, results reported in Figure 3d, implies a substantial and symmetrical drop in the binding energy, which goes from 42 meV for $F_{z}=0$ to 16 meV for $F_{z}=\pm 30$ kV/cm. It is clear that such a *z*-directed EF has the effect of increasing the average electron–impurity distance, which implies a drop in the Coulomb interaction and consequently in the binding energy. Speaking here of the degeneracies of different energy states, it is important to say that, for example, the EF breaks the triple degeneracy of the first excited state. It gives rise to a doubly degenerated energy state that comes from states with $p_{x}$ and $p_{y}$ symmetries which respond in the same way to the *z*-directed EF, and to a $p_{z}$ non-degenerate state whose two antinodes are located along the *z*-axis which responds differently to the *z*-directed EF. It is interesting to compare Figure 3c,d and to note that, in the absence of the impurity, the first excited state is doubly degenerate and can only be populated by optical transitions that involve linear polarization of the incident radiation along the *x*- or *y*-directions. In the case of Figure 3d, the presence of the impurity causes the first excited state to come from the second solution with l=0, which leads to $p_{z}$-like symmetry, and the state can be populated by absorption of resonant radiation with linear polarization along the *z*-axis. The degree of degeneracy of the states in Figure 3d are the same as in Figure 3c, given the central character of the impurity confinement.

In Figure 4 the lowest confined electron states in a CdS/ZnS spherical QD as functions of the MF are presented. Results are without, Figure 4a,c, and with, Figure 4b,d, on-center donor impurity, with $R_{1}=20$ nm and $R_{2}=30$ nm, for several values of the *l*-quantum number. Calculations are without EF, Figure 4a,b, whereas in Figure 4c,d the results are for $F_{r}=-20$ kV/cm central EF. From Figure 4a,c, it is observed that the MF breaks the degeneracy of states with the same absolute value of the *l*-quantum number. States with negative *l*-value initially show a decreasing character with the MF until they reach a minimum, and then they grow. States with positive *l*-value are always increasing in energy with the MF. From the same two figures, it is observed that the first and second excited states have degeneracies of order three and five, respectively, at zero MF, as it had been stated in the discussions of Figure 2 and Figure 3. Figure 4a shows that the fourth excited state corresponds to the fourth solution for l=0. In addition, from the same figure, it can be seen that regardless of the MF, the ground state retains the *s*-like symmetry (l=0). In Figure 4c, where a repulsive potential associated with the central EF is present, the system presents oscillations of the ground state. That is, for B<10 T, the ground state corresponds to the first solution with l=0, while for 10T<B<19 T it corresponds to l=−1. For 19T<B<20 T the ground state corresponds to l=−2. This is a typical behavior of confined electrons in three-dimensional core/shell systems or in two-dimensional quantum rings. It is interesting to note that in the presence of the central EF, the fifth solution with l=0 becomes the second non-degenerate state at zero MF. Figure 4b,d show the effects of the on-center impurity when the MF is present, in the absence/presence of the central EF, respectively. Again, as in previous results, it is observed that the energy levels are shifted towards lower values given the attractive nature of the impurity Coulomb potential. By comparing Figure 4a,b, one may observe that the impurity effect is definitely more significant on the states with l=0, particularly on the ground state. The second non-degenerate state, which in the absence of the impurity corresponds to the third excited state, in the presence of the impurity becomes the second excited state, with energy quite close to the first triply degenerate excited state at zero MF. It is evident from Figure 4b that the impurity does not change the symmetry of the ground state over the entire range of calculated MF. The situation is different when the central EF is applied. Note, in Figure 4d, that the larger impurity effects on states with l=0 imply the disappearance of the ground state oscillations shown in Figure 4c. This will cause an oscillation behavior of the ground state binding energy. For 0<B<10 T, the binding energy will be obtained from the difference between two states with l=0; in the range 10T<B<19 T, the binding energy will be given by the difference between a state with l=−1 and another one with l=0; finally, for 19T<B<20 T, the binding energy is given by the difference between a state with l=−2 and another one with l=0. There will be important implications on the optical properties of the system, because the WF symmetries radically affect the selection rules when the system is excited by radiations with a particular polarization. Taking into account the energy difference between the first two solutions with l=0, between the panels of Figure 4a,b there would be a change in energy of 2.61 meV when going from zero to 20 T, while in the case of the panels of Figure 4c,d, this difference becomes 3.87 meV.

In Figure 5, we present the same type of results as those reported in Figure 4, but this time with the EF applied along the *z*-direction. The results in Figure 5a,b are the same as those presented in Figure 4a,b. We have brought those results into this new figure for comparison purposes and to facilitate analysis. Clearly, the presence of the EF along the *z*-direction breaks the symmetry of the system, as it resulted from the discussion of Figure 3c,d. In this case, a *z*-directed EF gives rise to an electron confined in the region close to z=+R1, with the appearance of a quasi-conical QD, with maximum probability density located along the *z*-axis. This explains why the presence of the MF generates a marginal change of 2.3 meV on the first state with l=0 when MF increases from zero to 20 T. In addition, this type of confinement also justifies the disappearance of the ground-state oscillations that had been observed in Figure 4c. Another interesting behavior emerges by comparing the ground state binding energy results obtained from Figure 4c,d with those from Figure 5c,d. In the presence of the central (*z*-directed) EF, the binding energy goes from 25.72 meV (26.421 meV) at zero MF to 29.59 meV (27.904 meV) at B=20 T, so with a binding energy change of 3.87 meV (1.483 meV). It means that the more confined the system is along the *z*-axis, the lesser the effects of the MF are.

Figure 6, Figure 7, Figure 8 and Figure 9 present the squared matrix elements for transitions from the ground state to the first fourteen excited states (|1〉→|n〉, with n=2.3,…,15) of a confined electron in a CdS/ZnS spherical QD with l=0. The results are presented following the same scheme of energy as in Figure 2, Figure 3, Figure 4 and Figure 5, respectively. For example, in Figure 6 the results depend on the $R_{1}$-inner radius, without, Figure 6a, and with, Figure 6b, on-center donor impurity, for $R_{2}=30$ nm and without EF and MF effects. The analysis of all the figures goes through understanding the symmetries of the involved WFs. We are going to analyze some of the panels of the figures shown, analyzing the most relevant features or some of the most significant ones. Taking into account that a *z*-polarized linear incident radiation will be used to excite the optical transitions from the lowest state with l=0, we have to observe that any transition to final states with l≠0 will be forbidden. This fact is due to the null value of the xy-components in the integrals of the transition matrix elements. That restricts us to study transitions to final states with l=0. All states with l=0 are even functions concerning the changes x→−x and y→−y. This implies that in all cases the functions that appear in the dipole moments of the *x* and *y* coordinates are even with non-zero results for the integration.

Concerning Figure 6 we are going to focus our attention on $M_{1-2}^{2}$ in the absence/ presence of the impurity. In Figure 6a, the systematic increase of $M_{1-2}^{2}$ is due to the fact that increasing the QD $R_{1}$-radius, increases the space region where the |1〉 and |2〉 WFs are simultaneously extended. The two lobes of the |2〉 state systematically move away from the QD center, thus increasing $M_{1-2}^{2}$ since the overlap between the WFs of the two participating states is preserved. At all times, the spatial extent of the |1〉 and |2〉 states is approximately the same. When the impurity effects are present, a significant drop in $M_{1-2}^{2}$ is initially observed, whose value goes from 10 nm${}^{2}$ to 4.3 nm${}^{2}$ for $R_{1}=10$ nm. The reason for this is that the Coulomb interaction acts primarily on the |1〉 state, producing a significant decrease in the spatial extent of its WF. This fact reduces the overlap with the |2〉 state, producing the already observed decrease in $M_{1-2}^{2}$. As the $R_{1}$-radius increases, the |1〉 and |2〉 states extend in space and thus $M_{1-2}^{2}$ increases. In this range of increase of the transition matrix element, it is observed that the QD walls exert a confinement effect on the |1〉 state. The confinement effect on the ground state vanishes and the only WF that continues to grow in space when $R_{1}$ grows is that of the excited state. This results in a decrease in the overlap between the two states that participate in the dipole matrix element and consequently in a systematic fall in $M_{1-2}^{2}$, as seen in Figure 6b for $R_{1}>13$ nm.

In Figure 7a, we focus on the analysis of $M_{1-2}^{2}$, which represents a transition between an initial state with *s*-like symmetry and a final state with $2\;p_{z}$-like symmetry. The negative central EF generates a core/shell type confinement with the WFs located mainly in the region near the dot-barrier interface. In that case, $M_{1-2}^{2}$ takes a maximum value of 70 nm${}^{2}$. As the field approaches zero with negative values, the repulsive and central effect on the WFs is reduced.

They approach the central region of the QD, but preserving their high overlap value. This, then reduces the $M_{1-2}^{2}$ value, which approaches to 39 nm${}^{2}$ as $F_{r}\to 0^{-}$. When passing $F_{r}$ to the positive range values, a compression of the WFs towards the QD center appears, also preserving a high overlap value between the two states. This may be understood as a QD of lower size, which implies that for $F_{r}=30$ kV/cm the transition matrix element reaches the minimum value of 15.5 nm${}^{2}$. Analyzing the problem of the *z*-directed EF without impurity, Figure 7c, particularly for $M_{1-2}^{2}$, a systematic and symmetric decrease with the magnitude of the EF is observed. For $F_{z}=0$, obviously the results coincide with those presented in Figure 7a when $F_{r}=0$. For $F_{z}=0$, both WFs are distributed with their maximum extent around the QD center; therefore, $M_{1-2}^{2}$ takes its maximum value. When turning on the *z*-directed EF, there is a notable shift of the ground state WF along the *z*-direction while for the excited state, as it is more extended in space, the changes in its positions are less significant. This leads to a decrease in overlap between WFs and consequently to a decrease in $M_{1-2}^{2}$. By increasing the magnitude of $F_{z}$, the excited state starts to lose one of its lobes, with the other one being strongly localized towards the $z=\pm R_{1}$ region, depending on the $F_{z}$ sign. In that case, the ground state is located in the same region where the excited state concentrates its lobe, but it is more extended. This again implies a decrease in the overlap of the WFs and so $M_{1-2}^{2}$ continues to decrease. The analysis shows that it is necessary to have exact information about the WFs and their symmetries in order to understand the matrix elements behavior. Figure 7c reflects in $M_{1-3}^{2}$ and $M_{1-2}^{2}$ the symmetry change of the WFs that comes from the crossing of the second and third states shown in Figure 3b and which occurs close to $F_{z}=-7.5$ kV/cm.

Figure 8a–c show the numerical findings of $M_{1-2}^{2}$ as a function of the MF in the absence/presence of the central and *z*-directed EF. By considering only the MF, a variation of $M_{1-2}^{2}$ can be seen from 39.6 nm${}^{2}$ for B=0 up to 44.4 nm${}^{2}$ for B=20 T. Again, the ground state is the most sensitive to the presence of the MF. At zero MF, its geometry is spherical, becoming elliptical as *B* grows. This implies a displacement of the probability density of the ground state towards the regions where the excited state has its two lobes, giving rise to an increase in the overlap and consequently to an increase in the transition matrix element, as seen in Figure 8a. The combination of the negative central EF and the MF generates important changes on the ground state WF. First of all, the negative central EF generates a core/shell configuration as discussed above. When turning on the MF, a constriction of the WF occurs towards the central region of the QD, generating two lobes of the same sign located on the *z*-axis. The two lobes of the ground state coincide with the location of the two opposite sign lobes of the first excited state and the WFs overlap is reinforced as the MF increases. This explains the significant change in the transition matrix element in Figure 8c from 62.2 nm${}^{2}$ for B=0 up to 99.9 nm${}^{2}$ for B=20 T. Since the presence of the impurity in the center concentrates the ground state WF in its neighborhood, it is difficult to generate changes of its extent by MF effects. This explains the slight variations suffered by the matrix elements in Figure 8b,d, which only come from small changes in the excited state. It is important to say that the effects of the impurity on the ground state are dominant over those that can be induced by the presence of the negative central EF. In Figure 8d, the symmetry change in the WFs of the first two excited states can be seen, which is reflected in the complementary behavior of the matrix elements. As stated before, the negative *z*-directed EF concentrates both the ground state and the first excited state near the *z*-axis in the region $z=+R_{1}$. The presence of an MF hardly affects the shapes of the WFs since they are strongly localized from the start. This explains why in Figure 9c the changes in the matrix elements with the MF are marginal.

Next, we will make use of the results presented in previous figures to analyze the optical absorption coefficient (upper panels of Figure 10, Figure 11 and Figure 12) and the relative changes in the refractive index coefficient (lower panels of Figure 10, Figure 11 and Figure 12) as functions of the *z*-polarized incident photon energy. The results are presented in the absence (presence) of the impurity in the left (right) hand side panels of Figure 10, Figure 11 and Figure 12. In Figure 10, three values of the QD radius are considered according to the results in Figure 2 and Figure 6. In Figure 11, three values of the MF are considered according to the results in Figure 4a,b and Figure 8a,b. In Figure 12, three values of the MF combined with a negative central EF are considered according to the results in Figure 4c,d and Figure 8c,d. The results on the curves have been shifted along the vertical axis for clarity. In this sense, the vertical scale of each panel corresponds to the magnitude of the coefficient shown in the lower curve of each graph (red curve in each panel). (See the online version to identify the colors of the curves).

In Figure 10, where only the |1〉→|2〉 transition is considered, there is a drop in the magnitude of the resonant peak in the total absorption coefficient (α) which comes from the decrease in the transition energy as the radius of the QD increases, as can be seen from Figure 2. This occurs despite the fact that $M_{1-2}^{2}$ increases or decreases with the radius of the QD, as seen in Figure 6a,b, respectively. The magnitude of the resonant peak in the α-coefficient is dominated by the magnitude of the transition energy. In Figure 10c,d, it can be seen that in the absence (or presence) of the impurity, the magnitude of the relative changes in the refractive index coefficient (Δn) obeys the behavior of $M_{1-2}^{2}$. That is, when $M_{1-2}^{2}$ increases with the radius, as seen in Figure 6a, the Δn-coefficient also increases, as can be seen in Figure 6c. The drop in the magnitude of the resonant peaks in Figure 10d is in agreement with the drop in $M_{1-2}^{2}$ that is presented in Figure 6b for the three considered values of the radius. For $R_{1}=25$ nm, the important effect that the third-order correction can have on the α-coefficient is observed. This is evidenced by the strong suppression that occurs on the central peak. Of course, the third-order corrections in the two optical coefficients shown in Figure 10 are proportional to the intensity of the incident radiation. The intensity must be limited in this model as to avoid negative values in the total optical absorption coefficient, a situation that lacks any physical meaning.

In Figure 11, where three values of the MF are considered for a fixed geometry, without the EF effects, the points to be highlighted are: (i) again we concentrate on the |1〉→|2〉 transition, which is due to the dominant character shown by $M_{1-2}^{2}$ in Figure 8a,b; (ii) for the absorption coefficients and changes in the refractive index coefficient, there are no important variations in the magnitude and position of the resonant peaks as the MF changes. This is explainable by two reasons: (1) The variation of the transition energy between the two lowest states with l=0 in Figure 4 is very small. The slight shift towards higher energies (blue shift) in the resonant peaks of Figure 11a, or in the zeros of Figure 11c, is fundamentally associated with the increase in energy of the second state with l=0 in Figure 4b. (2) The variation presented by $M_{1-2}^{2}$ in Figure 8a,b does not exceed 16% throughout the range of considered MF. From Equation (Equation 3), it is evident that the first-order correction for the $\alpha^{\left(1\right)}$-coefficient is proportional to $E_{12}\;M_{1-2}^{2}$ and from Equation (Equation 6) we conclude that the $\Delta n^{\left(1\right)}$-coefficient is proportional to $M_{1-2}^{2}$. Ignoring the third-order corrections, which can be controlled by the intensity of the incident radiation, the discussion above justifies the few variations of the resonances in magnitude and position on the energy scale.

In Figure 12, we present the results concerning the optical coefficients depending on the MF, adding to the model the effect of a negative central EF. The results in this figure should be compared with those in Figure 11 where only the effect of the MF is considered. In the presence of the impurity, Figure 12b,d, the results on the magnitude of the resonant peaks do not differ much from those in Figure 11b,d. The shift towards lower transition energies, which is inferred from the difference in energy between the two lowest states with l=0 in Figure 4d, is compensated, although not completely, by the increase in $M_{1-2}^{2}$ and $M_{1-3}^{2}$ in Figure 8d. Note that, in general, the magnitudes of the resonant peaks in the absorption coefficient show a drop of 28% when the central EF is considered (in accord with the results in Figure 11b and Figure 12b). By comparing Figure 11d and Figure 12d, it is observed that the central EF generates an increase in the resonances of the Δn-coefficient. This is due to the higher value of the $M_{1-2}^{2}$ and $M_{1-3}^{2}$ terms in Figure 8d with respect to $M_{1-2}^{2}$ in Figure 8b. Comparing the results in Figure 4a,c shows that the central EF decreases the transition energy between the two lowest states with l=0, a situation that is accentuated as the MF increases. Taking into account that the magnitude of the $\alpha^{\left(1\right)}$-coefficient is proportional to $E_{12}$ (or $E_{13}$, depending on the MF), one could observe why in the curve for B=20 T of Figure 12a there are domains of photon energy where the absorption coefficient becomes negative. This is an indication that the intensity of the incident radiation in the third order correction model exceeds the physically acceptable limit. The closer the transition energy is to zero, the lower the intensity of the incident radiation must be in order to avoid the bleaching effect on the optical absorption coefficient.

A strong built-in EF in the order of MV/cm due to piezoelectricity has been found in wurtzite nitride heterostructures [57]. The electronic states in In${}_{x}$Ga${}_{1-x}$N/GaN core/shell QDs considering built-in EF have been calculated with the FEM by Liu and Ban [58]. On this basis, the density-matrix approach, the ternary mixed crystal size, and the light intensity effects on optical absorption coefficient and refraction index coefficient between conduction sub-bands 1s−1p, 1p−1d, 1p−2s, 2s−2p in In${}_{x}$Ga${}_{1-x}$N/GaN core/shell QDs are discussed in detail. Clearly, the design of the problem corresponds to a QD with a central EF and that can be simulated through the model of ${\vec{F}}_{r}$ introduced in this article.

In this research, we have focused our interest on the electronic and impurity states in spherical QDs and the corresponding intraband transitions. In this sense, in our model we do not consider the hole states, nor the electrostatic electron–hole interaction, that is, excitonic states, and the corresponding interband absorption. In a further investigation, we may approach the interband optical properties for core/shell QDs by considering multiple hole states and the electron–hole pairs.

## 4. Conclusions

In conclusion, the ground state and the first fourteen excited states in CdS/ZnS spherical core/shell QD under the applied electric and MF and in the presence of an on-center donor impurity are investigated. The results show that all energy levels decrease as the internal radius of the QD increases. In the presence of an on-central impurity, the energy levels present a shift towards lower energies. In the *z*-directed EF case, a symmetric drop in energies is observed that even leads to negative energies for the higher *z*-directed EF regime. In the central EF case, we found the increasing and positive energies for the positive central EF and decreasing and negative energies for the negative central EF. The numerical findings show that the MF can break the degeneracy of states with the same *l*-value. The analysis results presented here indicate that information about the WFs and their symmetries is important for understanding the transition matrix elements behavior. The combination of the EF and MF generates significant changes on the WFs. Thus, the matrix elements greatly depend on the inner radius, the donor impurity, the EF, and MF. In addition, we have studied the optical absorption coefficient and the relative change of the refractive index coefficient, both of them for intraband transitions only. The magnitude of the resonant peak in the absorption coefficient can be dominated by the magnitude of the transition energy. In the presence of the central EF, the resonant peaks’ values of the absorption coefficient show a drop. It is observed that the central EF can generate an increase in the resonant magnitude of the Δn-coefficient. The results of this research can be suitable for potential applications in optoelectronic devices. Finally, we should mention that this article is predictive about the EF and MF effects on the optical and impurity properties in CdS/ZnS QDs. Therefore, we hope that the results presented here may stimulate experimentalists to investigate the optoelectronic properties of spherical CdS/ZnS QDs under EF and MF.

## Figures and Tables

**Figure 1 nanomaterials-12-04014-f001:**
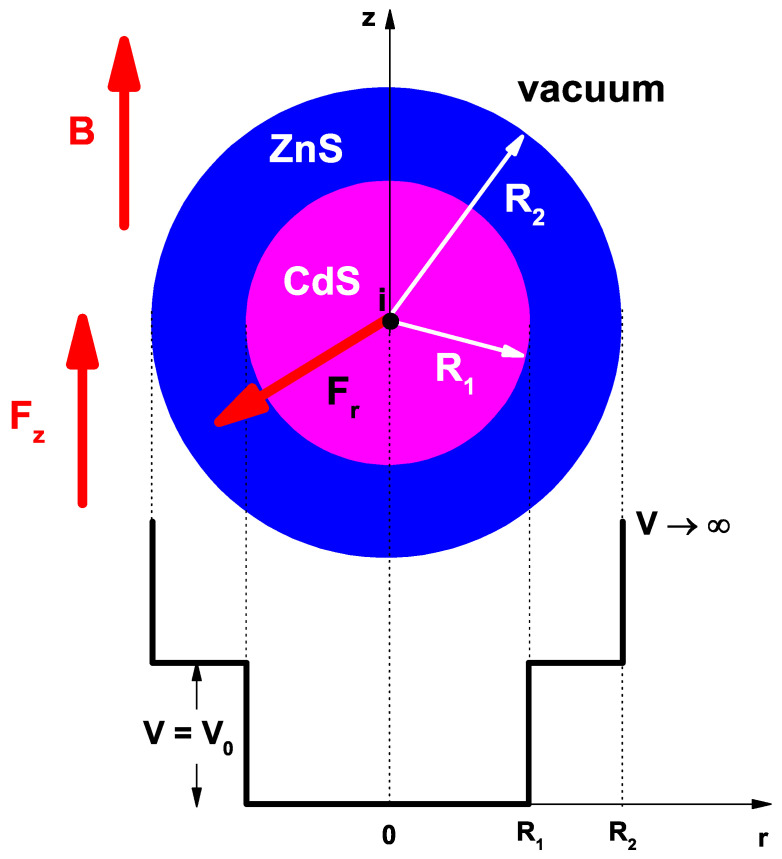
(color online) Pictorial view of the CdS/ZnS spherical QD studied in this work. The QD size, the *z*-directed applied EF and MF ($F_{z}$ and *B*), and the central EF ($F_{r}$) are indicated, together with the *r*-dependent confinement potential.

**Figure 2 nanomaterials-12-04014-f002:**
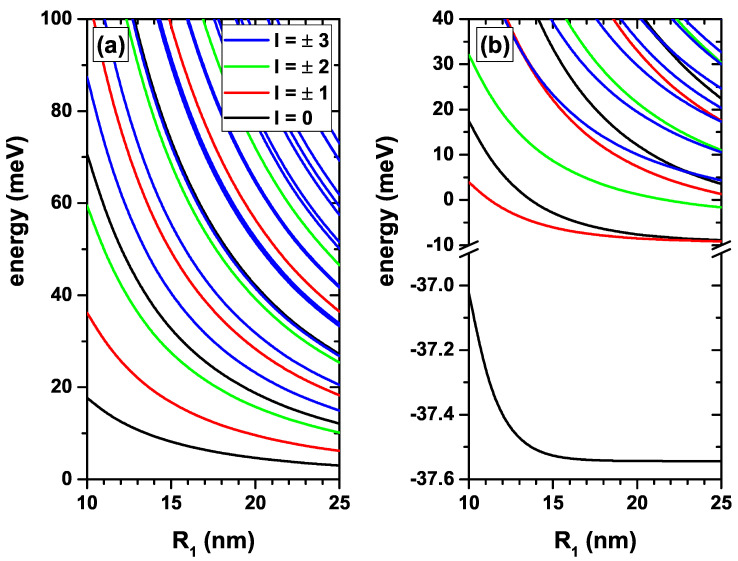
(color online) The lowest confined electron states in a CdS/ZnS spherical QD as functions of the $R_{1}$-inner radius. Results are without (**a**) and with (**b**) on-center donor impurity, with $R_{2}=30$ nm, for several values of the *l*-quantum number, and without EF and MF effects.

**Figure 3 nanomaterials-12-04014-f003:**
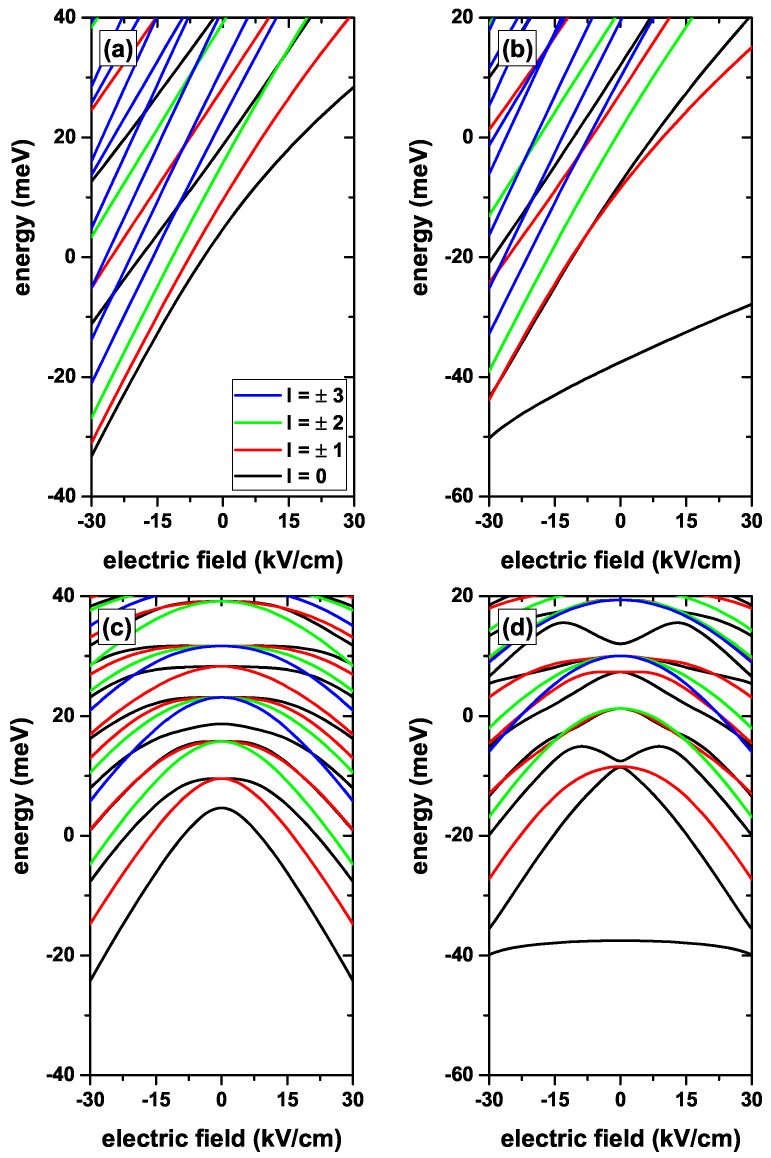
(color online) The lowest confined electron states in a CdS/ZnS spherical QD as functions of the central (**a**,**b**) and *z*-directed (**c**,**d**) EFs. Results are without (**a**,**c**) and with (**b**,**d**) on-center donor impurity, with $R_{1}=20$ nm and $R_{2}=30$ nm, for several values of the *l*-quantum number and without MF effects.

**Figure 4 nanomaterials-12-04014-f004:**
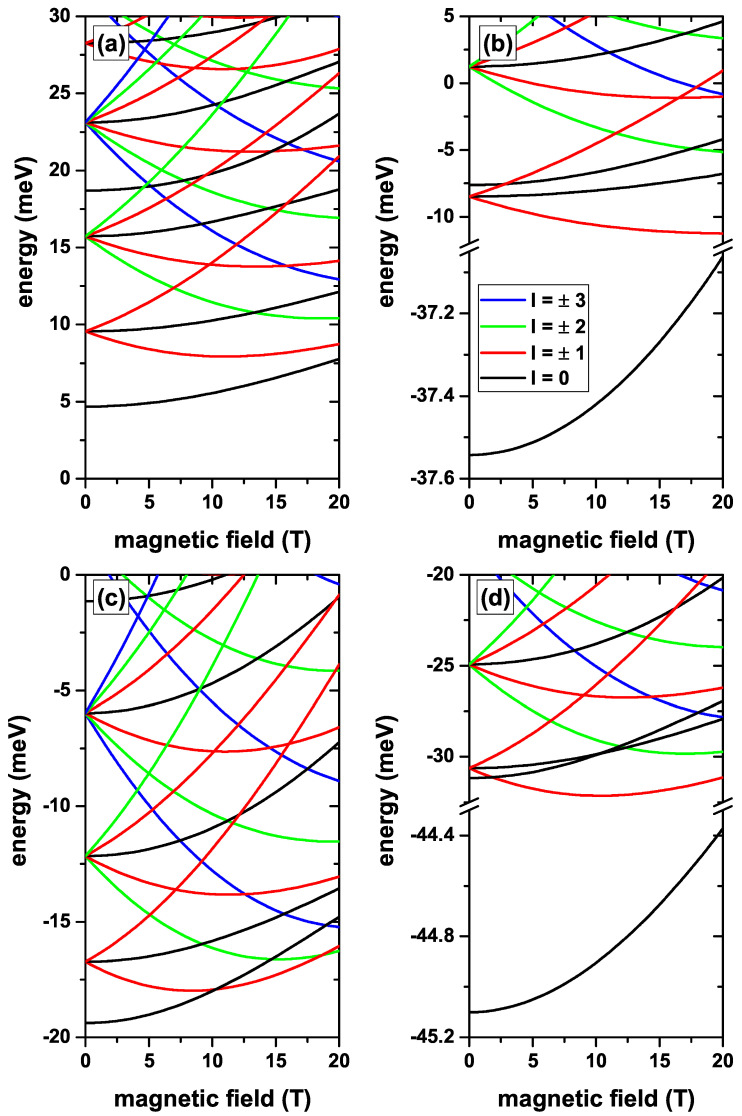
(color online) The lowest confined electron states in a CdS/ZnS spherical QD as functions of the MF. Results are without (**a**,**c**) and with (**b**,**d**) on-center donor impurity, with $R_{1}=20$ nm and $R_{2}=30$ nm, for several values of the *l*-quantum number. Calculations are with no EF in (**a**,**b**), whereas in (**c**,**d**) the results are for a central EF F=−20 kV/cm.

**Figure 5 nanomaterials-12-04014-f005:**
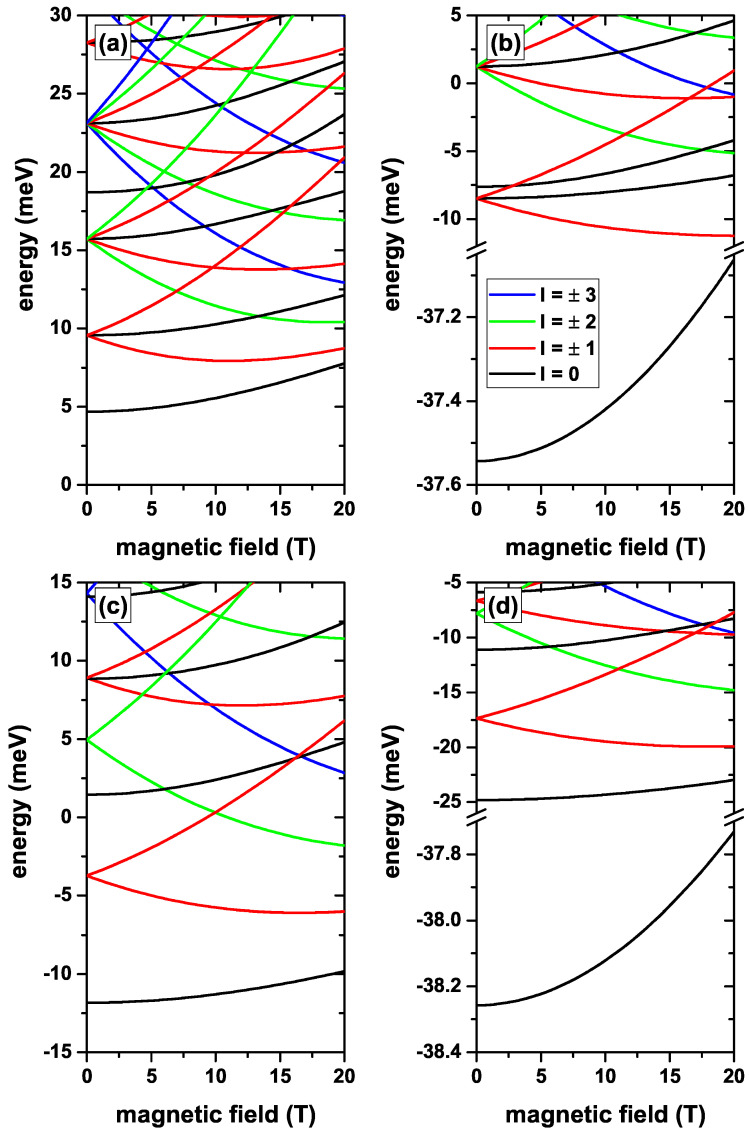
(color online) The lowest confined electron states in a CdS/ZnS spherical QD as functions of the MF. Results are without (**a**,**c**) and with (**b**,**d**) on-center donor impurity, with $R_{1}=20$ nm and $R_{2}=30$ nm, for several values of the *l*-quantum number. Calculations in (**a**,**b**) are for zero EF, whereas in (**c**,**d**) the results are for F=−20 kV/cm, *z*-directed EF.

**Figure 6 nanomaterials-12-04014-f006:**
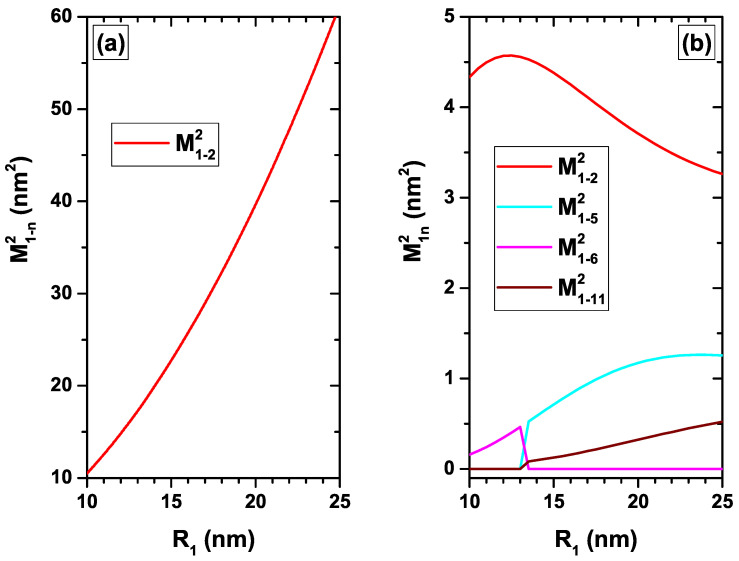
(color online) The squared matrix elements for transitions from the ground state to the first fourteen excited states with l=0 (|1〉→|n〉, with n=2,3,…,15) of a confined electron in a CdS/ZnS spherical QD as functions of the $R_{1}$-inner radius. Results are without (**a**) and with (**b**) on-center donor impurity, with $R_{2}=30$ nm, and without EF and MF effects.

**Figure 7 nanomaterials-12-04014-f007:**
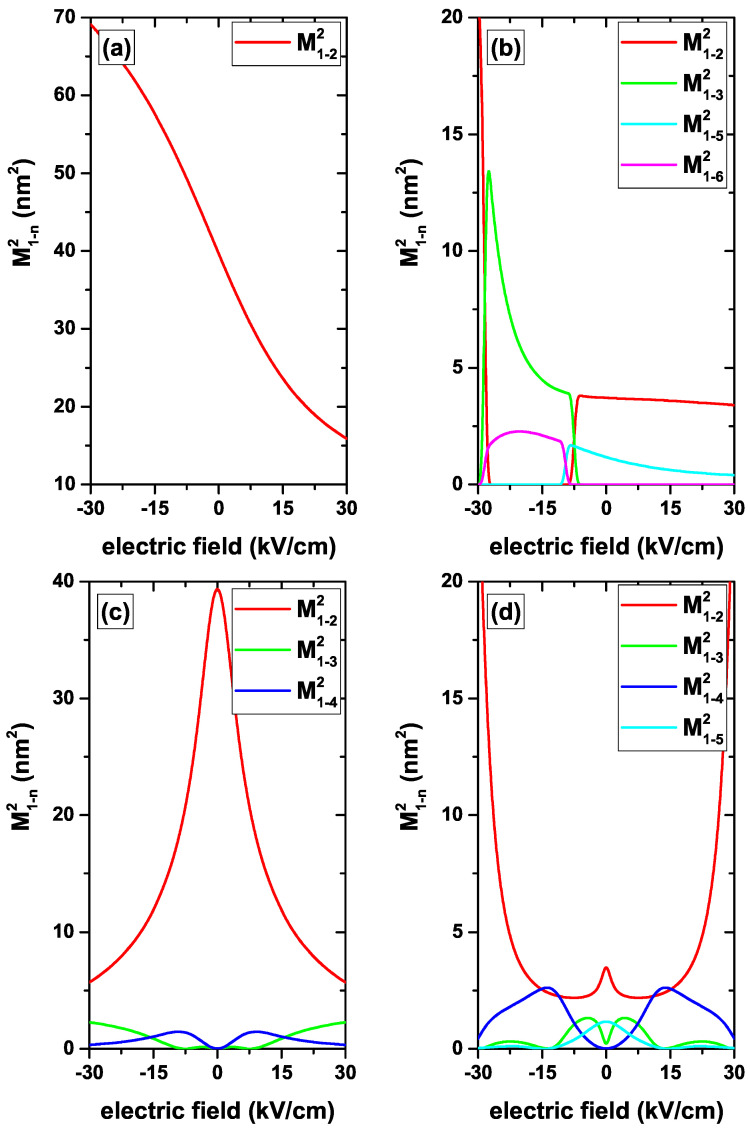
(color online) The squared matrix elements of the transitions from the ground state to the first fourteen excited states with l=0 (|1〉→|n〉, with n=2,3,...,15) for a confined electron in a CdS/ZnS spherical QD as functions of the central (**a**,**b**) and *z*-directed (**c**,**d**) EF. Results are without (**a**,**c**) and with (**b**,**d**) on-center donor impurity, with $R_{1}=20$ nm and $R_{2}=30$ nm.

**Figure 8 nanomaterials-12-04014-f008:**
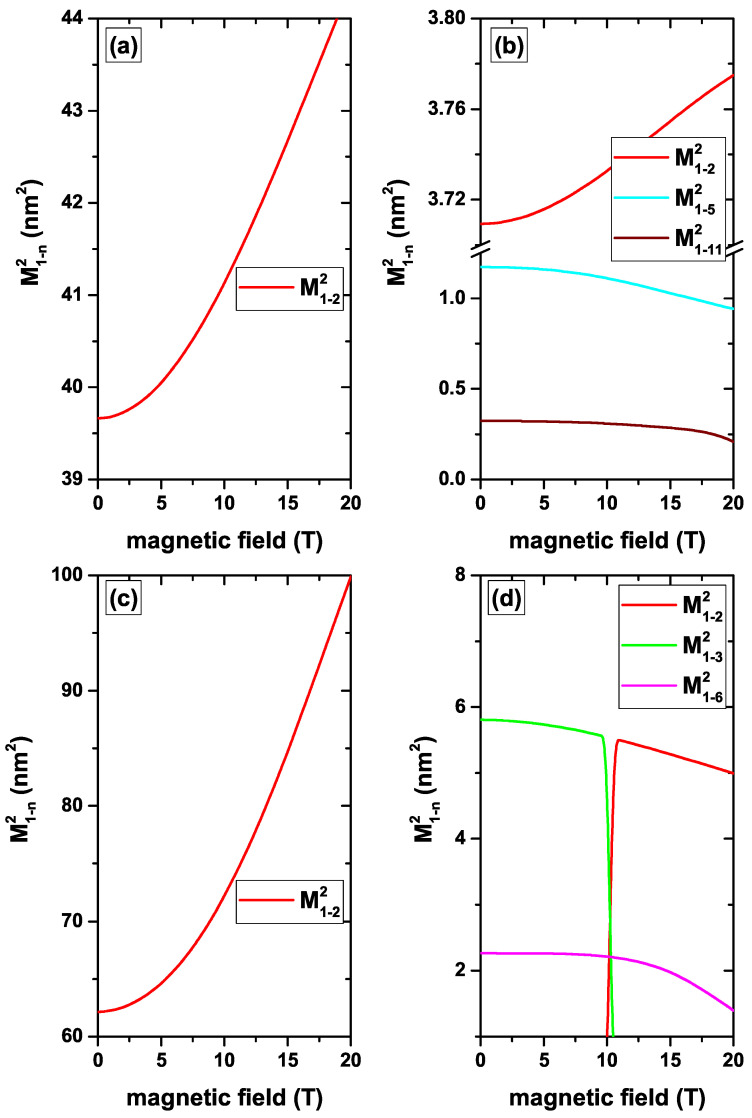
(color online) The squared matrix elements of the transitions from the ground state to the first fourteen excited states with l=0 (|1〉→|n〉, with n=2,3,…,15) for a confined electron in a CdS/ZnS spherical QD as functions of the MF. Results are without (**a**,**c**) and with (**b**,**d**) on-center donor impurity, with $R_{1}=20$ nm and $R_{2}=30$ nm. Calculations are for zero EF (**a**,**b**) whereas in (**c**,**d**) the results are for F=−20 kV/cm, central EF.

**Figure 9 nanomaterials-12-04014-f009:**
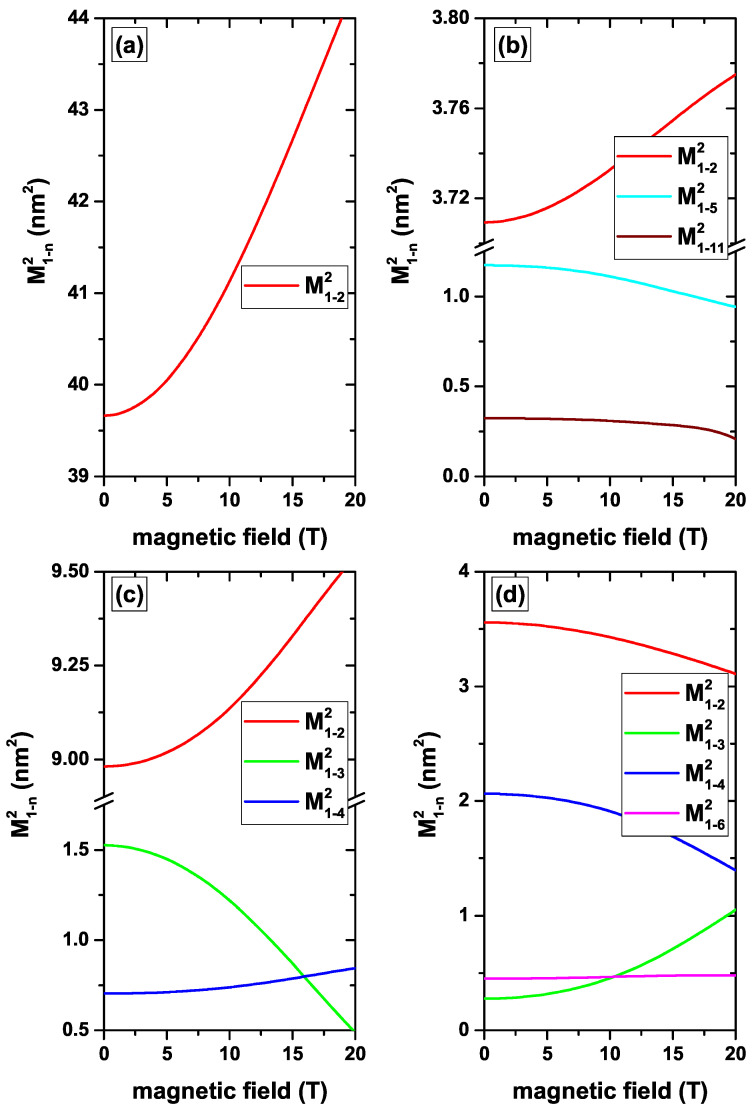
(color online) The squared matrix elements of the transitions from the ground state to the first fourteen excited states with l=0 (|1〉→|n〉, with n=2,3,…,15) for a confined electron in a CdS/ZnS spherical QD as functions of the MF. Results are without (**a**,**c**) and with (**b**,**d**) on-center donor impurity, with $R_{1}=20$ nm and $R_{2}=30$ nm. Calculations are for zero EF (**a**,**b**) whereas in (**c**,**d**) the results are for F=−20 kV/cm, *z*-directed EF.

**Figure 10 nanomaterials-12-04014-f010:**
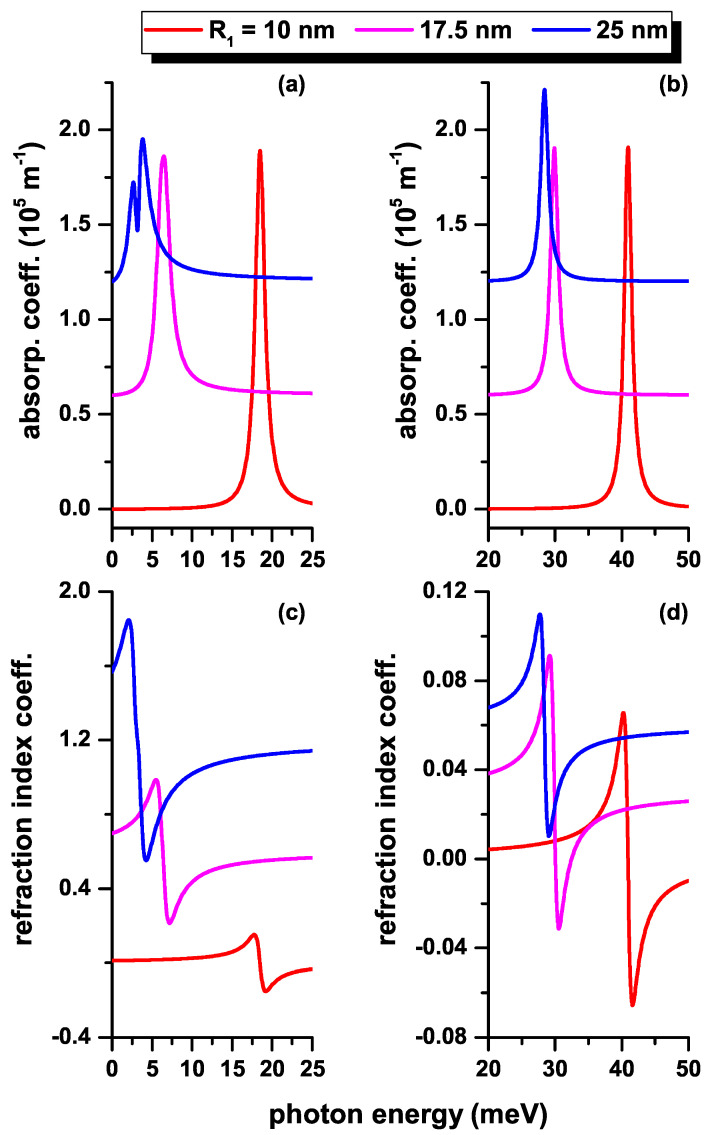
(color online) The total optical absorption coefficient (**a**,**b**) and the total relative changes of the refraction index coefficient (**c**,**d**) as functions of the incident photon energy in a CdS/ZnS spherical QD with several values of the $R_{1}$-inner radius. Results are without (**a**,**c**) and with (**b**,**d**) on-center donor impurity, with $R_{2}=30$ nm. Optical transitions are between the lowest confined l=0 states without electric and MF effects.

**Figure 11 nanomaterials-12-04014-f011:**
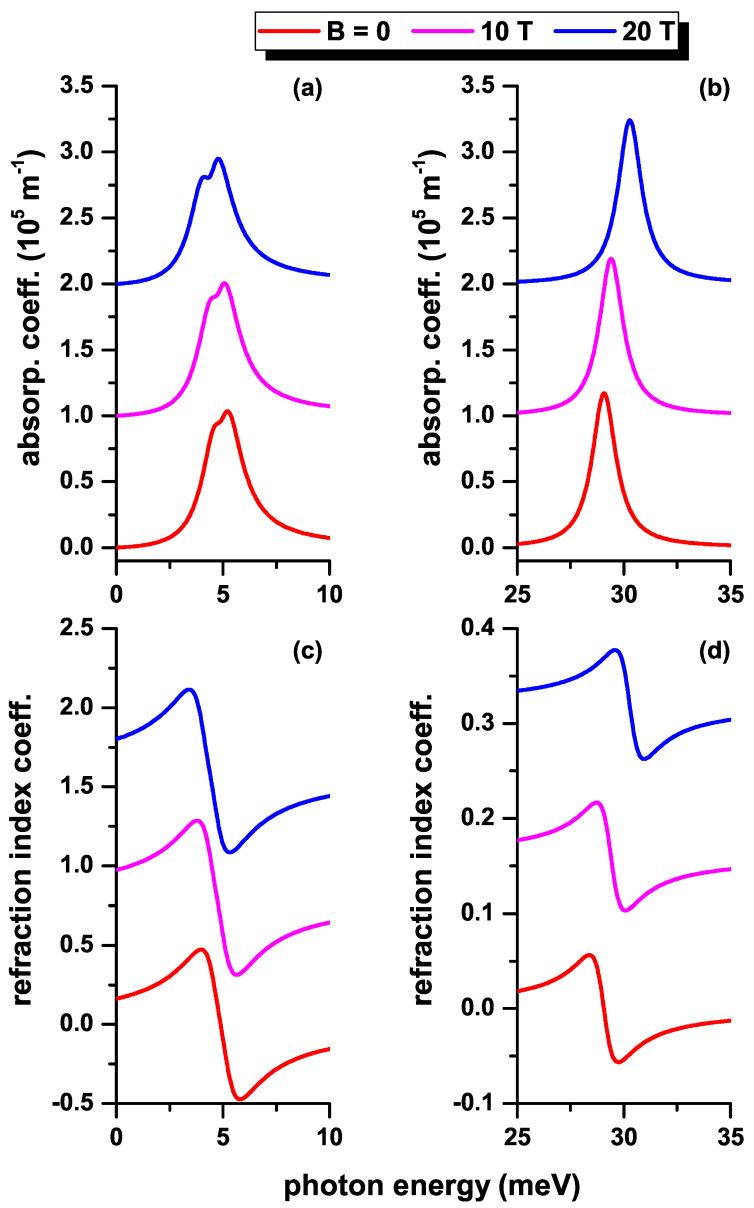
(color online) The total optical absorption coefficient (**a**,**b**) and the total relative changes of the refraction index coefficient (**c**,**d**) as functions of the incident photon energy in a CdS/ZnS spherical QD for several values of the MF. Results are without (**a**,**c**) and with (**b**,**d**) on-center donor impurity, with $R_{1}=20$ nm and $R_{2}=30$ nm. Optical transitions are between the lowest confined l=0 states without EF effects.

**Figure 12 nanomaterials-12-04014-f012:**
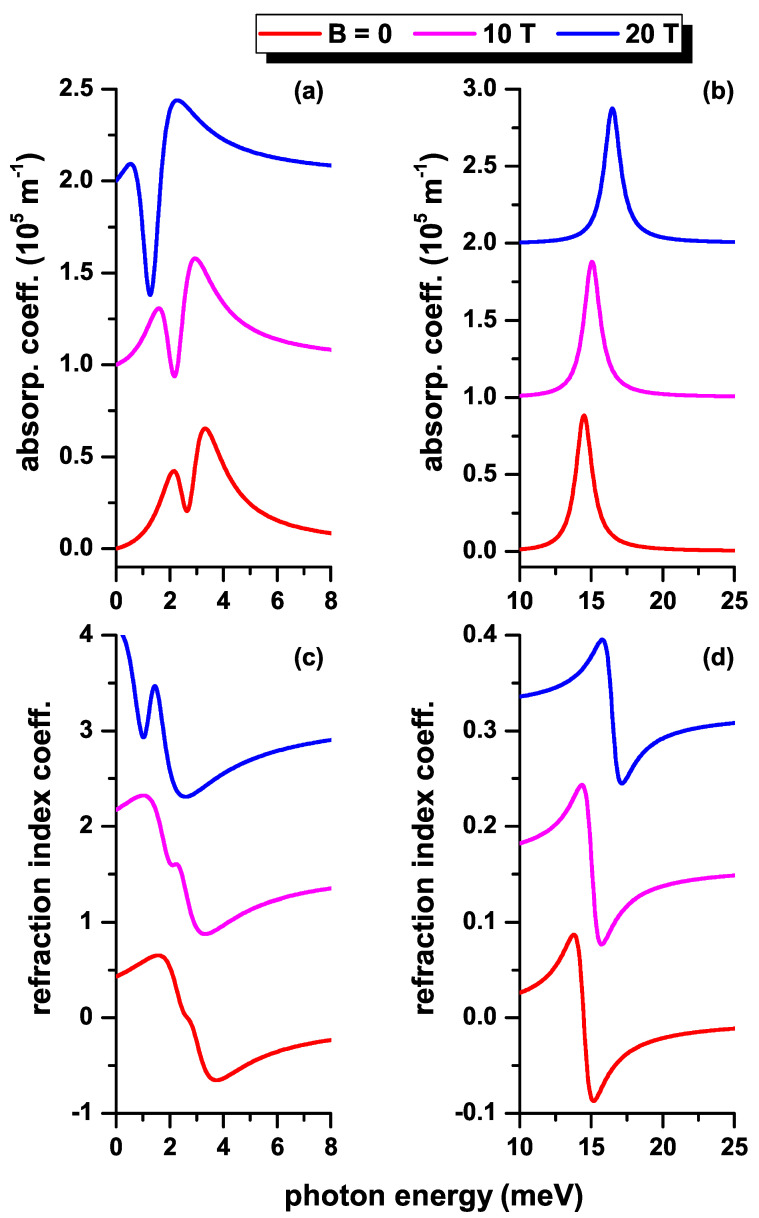
(color online) The total optical absorption coefficient (**a**,**b**) and the total relative changes of the refraction index coefficient (**c**,**d**) as functions of the incident photon energy in a CdS/ZnS spherical QD, for several values of the MF. Results are without (**a**,**c**) and with (**b**,**d**) on-center donor impurity, with $R_{1}=20$ nm and $R_{2}=30$ nm. Optical transitions are between the lowest confined l=0 states, considering a central EF F=−20 kV/cm.

## Data Availability

No new data were created or analyzed in this study. Data sharing is not applicable to this article.

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
