# Peer review of "Donor Impurity in CdS/ZnS Spherical Quantum Dots under Applied Electric and Magnetic Fields"

_nanomaterials, 2022, doi:10.3390/nano12224014_

Round 1

Reviewer 1 Report

The manuscript by Hasanirokh and coworkers presents a theoretical study of the properties of CdS/ZnS quantum dots with an impurity, under applied electric and magnetic fields. The authors use standard effective mass approximation and finite element method to determine the eigenfunctions and eigenenergies of the system and then compute the optical properties. It is claimed that the presence of impurity influences the optical properties and suppresses the oscillations caused by the electric field.

Firstly, the scope and potential of the presented study is not clear at all, neither is discussed in the paper. Secondly, the text is difficult to follow and in some cases it is incomprehensible. Language polishing by a native English speaker is in order. Also the presentation is confusing. For instance the usage of the terms "axial" and "radial" to describe the direction of electric and magnetic fields in a spherical geometry is somewhat misleading. Isn't the z-axis in Fig. 1 also "radial"? Could the authors clarify? Although mentioned that simultaneous application of electric and magnetic fields is considered the majority of the presented results are shown with only one of the two fields  "on". In the calculation of the absorption coefficients shouldn't one sum over the conduction and valence band states? Also, what about correlation and excitonic effects? Shouldn't they dominate the optical properties? In addition, it seems that some important relevant literature is missing (see e.g. doi: 10.3390/nano12010060). Finally, regarding novelty, standard methods are employed (effective mass approximation, finite elements method) and no new methodology is proposed.

In view of the above, I am afraid I cannot recommend the paper for publication in the present form. Should the authors would like to make amendments to the paper according to the comments, I would be willing to review their revised manuscript.

Author Response

Referee 1

The Referee:

The manuscript by Hasanirokh and coworkers presents a theoretical study of the properties of CdS/ZnS quantum dots with an impurity, under applied electric and magnetic fields. The authors use standard effective mass approximation and finite element method to determine the eigenfunctions and eigenenergies of the system and then compute the optical properties. It is claimed that the presence of impurity influences the optical properties and suppresses the oscillations caused by the electric field.

Our reply:

We want to present our most sincere thanks to the Referee for his/her report. The questions, comments, and suggestions of the Referee have helped us to identify the deficiencies in our manuscript and we consider that they have been a very important contribution to improve the presentation of the article and make it more attractive and interesting for readers.

The Referee:

Firstly, the scope and potential of the presented study is not clear at all, neither is discussed in the paper.

Our reply:

In the last paragraph of the Introduction section we added the following comment:

Despite the particular choice of CdS/ZnS QDs, the results discussed here are valid for many other material combinations that give rise to zero-dimensional structures with spherical symmetry, for example, GaAs/AlGaAs and InAs/GaAs QDs. Both the electric and magnetic fields, to be chosen along the $z$-axis, break the spherical symmetry of the problem while the version of the electric field directed radially from the QD center or toward the QD center will preserve such symmetry, even in the presence of a donor impurity located in the center of the system. Technologically, it is complicated to think about radial electric fields, however the study proposal here, unknown until now in the literature, can be perfectly used to simulate spherical structures with materials where the well-known built-in electric fields (induced by the discontinuities of the spontaneous and the piezoelectric polarizations at the boundaries of the QD structure) are present. The study is done in the effective mass and parabolic conduction band approximations. The effects of the effective mass and dielectric constant mismatch between the dot and barrier regions are included.

The Referee:

Secondly, the text is difficult to follow and in some cases it is incomprehensible. Language polishing by a native English speaker is in order.

Our reply:

We want to thank the Referee for his/her observation. We have carefully reviewed the manuscript for writing, spelling, and grammar issues, among others. We have polished the wording of the manuscript seeking greater clarity in the texts. Looking to make reading a little more personable.

The Referee:

Also the presentation is confusing. For instance the usage of the terms "axial" and "radial" to describe the direction of electric and magnetic fields in a spherical geometry is somewhat misleading. Isn't the z-axis in Fig. 1 also "radial"? Could the authors clarify?

Our reply:

With the following text in the Abstract section we clarify about the directions of the applied electric and magnetic fields.

The magnetic field is applied along the $z$-axis, with the donor impurity always located in the center of the quantum dot. In the case of the electric field, two situations have been considered: applied along the $z$-axis and applied in the radial direction (central electric field). For both cases of the considered electric field, the azimuthal symmetry (along the $z$-axis) is preserved

The caption of Fig. 1 has been improved.

Throughout the entire manuscript we have presented more clearly the two types of electric fields that have been studied.

The Referee:

Although mentioned that simultaneous application of electric and magnetic fields is considered the majority of the presented results are shown with only one of the two fields "on".

Our reply:

We thank the Referee for his/her observation. The Referee is absolutely right, in those cases where the effects of size and electric field are calculated (Figs. 2, 3, 6, 7, 10), we have considered the system in the absence of a magnetic field. In those figures where the magnetic field is considered as a variable (Figs. 4, 5, 8, 9, 11, 12), we have considered the presence of both central electric fields and electric fields along the z-direction. We have chosen this strategy taking into account the extreme difficulty that it would have to analyze the effects of each field if both the magnetic and the electric fields were included in all the results. We have tried to start from the most particular to move towards the most general. In the case of magnetic fields as a calculation variable, the presence of a central electric field has been interesting to the extent that it gives rise to core/shell type structures where, for example, the oscillations of the ground state are observed (ground state corresponding to different values ​​of the $l$-quantum number, depending on the value of the applied magnetic field). We ask the Referee to allow us to preserve the structure of the article, since including new figures would make the work too extensive, which would possibly limit a bit of interest for readers.

The Referee:

In the calculation of the absorption coefficients shouldn't one sum over the conduction and valence band states? Also, what about correlation and excitonic effects? Shouldn't they dominate the optical properties?

Our reply:

We want to thank the Referee for his/her observation and comment. In this research we have focused our interest on the electronic and impurity states in spherical quantum dots and the corresponding intraband transitions. In this sense, in our model we do not consider the hole states, nor the electrostatic electron-hole interaction, that is, excitonic states, and the corresponding interband optical absorption. In a Photoluminescence experiment, it is clear that excitonic structures should appear in the spectrum, with the presence of some peaks associated with impurity states. The Referee is right in his/her appreciation, in those cases of strong confinement, that is, quantum dots whose dimensions are less than the effective Bohr radius, multiple structures must be identified for interband transitions associated with the different confined electron states. To clarify this point, in the last parto f the Results and Discussion section of the revised manuscript, we have added the following text:

We have to stress that in this research we have focused our interest on the electronic and impurity states in spherical quantum dots and the corresponding intraband transitions. In this sense, in our model we do not consider the hole states, nor the electrostatic electron-hole interaction, that is, excitonic states, and the corresponding interband absorption. In a further investigation, which will be published elsewhere, we will study the interband optical properties for core/shell quantum dots considering multiple hole states and the electrostatic electron-hole interaction.

The Referee:

In addition, it seems that some important relevant literature is missing (see e.g. doi: 10.3390/nano12010060). 

Our reply:

We want to thank the Referee for drawing our attention to this Reference which we consider extremely appropriate for the elaboration of the state of the art of our article. We have included the suggested reference. In the second paragraph of the Introduction section we have added the following text:

Mkrtchyan \textit{et al.} have reported a theoretical study of effects of an external magnetic field on the interband and intraband optical properties of an asymmetric biconvex lens-shaped quantum dot \cite{Mkrtchyan}. In the theoretical model, the authors resort to the adiabatic approximation, which makes it possible to obtain analytical solutions for the eigenvalue equations in terms of sinusoidal and confluent hypergeometric functions. Leaving aside the effects of electron-hole interaction, the authors report the interband absorption coefficient, the photoluminescence coefficient, and the first and third order corrections of the optical absorption coefficient.

The Referee:

Finally, regarding novelty, standard methods are employed (effective mass approximation, finite elements method) and no new methodology is proposed.

Our reply:

We want to thank the Referee for his/her comment and appreciation. The Referee is right, in our model we make use of the effective mass approximation to write the Schrödinger equation. We have used the finite element method to solve the eigenvalue differential equation. Since in all cases the symmetry of the system around the z-axis is preserved, we have decided to work in cylindrical coordinates, with which it is only necessary to solve the eigenvalues  differential equation ​in the half of the structure projection in the xz-plane. In our numerical calculation, considering the presence of a divergence at r=0 that is associated with the electron-impurity Coulomb potential, we proceed to use spatial refinements of the meshes where the operators are discretized. The presence of the magnetic field and the electric field along the z-direction break the spherical symmetry of the problem, which from many points of view introduces strong numerical complications. We want to emphasize that many studies have been done using, for example, the variational method to introduce the impurity effects. The quantum dot problem in the presence of the two considered fields in fact no longer allows an analytical solution to be found to the eigenvalue problem in the absence of impurity. So, we believe that in this article we are presenting a set of results that could hardly be obtained with methods and approaches different from the one used here. Another strategy would be to use finite difference methods, including expansions of wave functions in orthonormal bases to proceed to a diagonalization calculation. Historically we have resorted to these other methods, but our experience has led us to lean towards the strategy used here.

We believe that the greatest novelty of our research lies in studying the effects of electric and magnetic fields together with the presence of an impurity in isolated quantum dots that, depending on the way in which the fields are introduced, can give rise to core/shell type structures.

The Referee:

In view of the above, I am afraid I cannot recommend the paper for publication in the present form. Should the authors would like to make amendments to the paper according to the comments, I would be willing to review their revised manuscript.

Our reply:

Again, we want to thank the Referee for the careful reading and review of our manuscript. His/her suggestions and comments have helped us substantially improve the quality of our article. We hope that the Referee finds our responses and changes to the revised version satisfactory and that he/she considers our article suitable for publication in the journal Nanomaterials.

Reviewer 2 Report

The paper presents a numerical study on the optical properties of CdS/ZnS quantum dots, and on the influence that impurities and externally applied fields have on those properties. 

The paper is quite clearly written, it is coherent with the chosen review and  the results are clearly explained. I think it is suitable for publication, provided that the following points are addressed: 

1) I found that the scientific context of the paper could have been better explained. Just to mention a few questions that came into my mind reading it:

Are the obtained results general in the context of core-shell nanosystems or are they specific of the CdS/ZnS NP as the title suggests? In my opinion the findings should be quite general. So it would be better explain that CdS/ZnS has been chosen as an example to put some numbers, but that the method is general and can be applied to other systems.

Are there experimental data that can be compared to the findings obtained by the authors? How do the results they obtain compare to existing literature?

In the paper the effect of a radial electric field is considered. While the effect of a z-oriented electric field and of a z-oriented magnetic field can be understood in terms of a uniform external field applied to a nanoparticle collection, it is less clear to me in which context a radially directed, uniform electric field is present. Can the authors elaborate a little on the interest of this configuration and in which practical cases it is encountered?

Please add some examples of typical donor impurities used in CdS/ZnS nanoparticles with some reference. 

Second: the English of the paper needs a revision: several sentences do not feel correct. To mention a few: 

"The optical absorption coefficients under electric and MFs and considering impurity are enhanced the optical and electronic properties of the system" (page 2)

"The DMEs that do not appear is because they are zero." (caption of Fig. 6 and other figures)

"the changes in its position are slight" (page 13)

etc.

Author Response

Referee 2

The Referee:

The paper presents a numerical study on the optical properties of CdS/ZnS quantum dots, and on the influence that impurities and externally applied fields have on those properties.

The paper is quite clearly written, it is coherent with the chosen review and the results are clearly explained. I think it is suitable for publication, provided that the following points are addressed: 

Our reply:

We want to present our most sincere thanks to the Referee for his/her report. The questions, comments, and suggestions of the Referee have helped us to identify the deficiencies in our manuscript and we consider that they have been a very important contribution to improve the presentation of the article and make it more attractive and interesting for readers.

The Referee:

1) I found that the scientific context of the paper could have been better explained. Just to mention a few questions that came into my mind reading it:

Are the obtained results general in the context of core-shell nanosystems or are they specific of the CdS/ZnS NP as the title suggests? In my opinion the findings should be quite general. So it would be better explain that CdS/ZnS has been chosen as an example to put some numbers, but that the method is general and can be applied to other systems.

Our reply:

We completely agree with the Referee and appreciate his/her observation. In the last part of the Theoretical Model section we have added the following text:

At this point, it is very important to highlight that the results that we present below are particularly for CdS/ZnS QDs, but it must be taken into account that they are of a general nature and can be applied to other combinations of materials in the dot and barrier regions. Changes in the materials essentially imply changes in the effective mass parameters consequently in the magnitude of the effective Bohr Radius and of the effective Rydberg, which finally translates into changes in the magnitudes of the energies of the confined states, of the dipole matrix elements for intraband transitions, and impurity related binding energy.”

The Referee:

Are there experimental data that can be compared to the findings obtained by the authors? How do the results they obtain compare to existing literature?

Our reply:

We thank the Referee for this question, which we consider to be of the highest relevance. In general, it has been shown that the effective mass approximation is an excellent choice to describe the Physics of systems confined in the three directions of space, particularly in the case of QDs with spherical symmetry. In that sense, the results reported here give a good account of experimental reports on CdS/ZnS QDs in the absence of external fields. Making a careful review of the literature, we have not found experimental reports about the effects of electric and magnetic fields on optical and impurity properties in CdS/ZnS QDs. This leads us to say that this article is predictive in nature and that we hope that the results presented here will stimulate experimentalists to study the properties of these confined 3D systems. At the end of the Conclusions section we have added the following text:

Finally, we must mention that this article is predictive about the electric and magnetic field effects on the optical and impurity properties in CdS/ZnS QDs. Therefore, we hope that the results presented here will stimulate experimentalists to study the optoelectronic properties of spherical CdS/ZnS QDs under externally applied electric and magnetic field.

The Referee:

In the paper the effect of a radial electric field is considered. While the effect of a z-oriented electric field and of a z-oriented magnetic field can be understood in terms of a uniform external field applied to a nanoparticle collection, it is less clear to me in which context a radially directed, uniform electric field is present. Can the authors elaborate a little on the interest of this configuration and in which practical cases it is encountered?

Our reply:

At the end of the Results and Discussion section we added the following comment with its corresponding references:

A strong built-in electric field in the order of MV/cm due to piezoelectricity and spontaneous has been found in wurtzite nitride heterostructures \cite{Shi_2003}. The electronic states in In$_x$Ga$_{1-x}$N/GaN core/shell QDs considering built-in electric field have been calculated by the means of FEM firstly by Liu and Ban \cite{Liu_2017}. On this basis, applying the density-matrix approach, the ternary mixed crystal, size and light intensity effects on optical absorption coefficient and and refraction index coefficient between inter sub-conduction bands $1s-1p$, $1p-1d$, $1p-2s$, $2s-2p$ in In$_x$Ga$_{1-x}$N/GaN core/shell QDs were discussed in detail. Clearly, the design of the problem corresponds to a QD with a central electric field and that can be simulated through the model of $\vec{F}_r$ introduced in this article.

The Referee:

Please add some examples of typical donor impurities used in CdS/ZnS nanoparticles with some reference.

Our reply:

In the Instroduction section we added the following paragraph with its corresponding references:

The results obtained for CdS films chemically deposited from solutions containing different impurities known as donor type dopants for CdS (chlorine, iodine, boron and indium) have been reported by Altosaar \textit{et al.} \cite{Altosaar_2005}. In CdS, Cd and S vacancies result in the creation of various acceptor and donor states within the forbidden energy gap. This will cause the transition of charge carriers through defect states to the valence band, following by the broadening of photoluminescence spectra as well as deviation from defect-free ideal bandgap energy value of semiconductor \cite{Khatter_2021}. When fluorine atoms are incorporated into de CdS lattice, they can act as donor impurities within the CdS bandgap, these levels will eventually induce a CdS bandgap reduction for increasing fluorine concentration in the chemical bath \cite{Nieto-Zepeda}.

The Referee:

Second: the English of the paper needs a revision: several sentences do not feel correct. To mention a few:

"The optical absorption coefficients under electric and MFs and considering impurity are enhanced the optical and electronic properties of the system" (page 2)

"The DMEs that do not appear is because they are zero." (caption of Fig. 6 and other figures)

"the changes in its position are slight" (page 13)

etc.

Our reply:

We want to thank the Referee for his/her observation. We have carefully reviewed the manuscript for writing, spelling, and grammar issues, among others. We have polished the wording of the manuscript seeking greater clarity in the texts. Looking to make reading a little more personable.

Again, we want to thank the Referee for the careful reading and review of our manuscript. His/her suggestions and comments have helped us substantially improve the quality of our article. We hope that the Referee finds our responses and changes to the revised version satisfactory and that he/she considers our article suitable for publication in the journal Nanomaterials.
